# Neurobiology of Coughing in Children

**DOI:** 10.3390/jcm12237285

**Published:** 2023-11-24

**Authors:** Stuart B. Mazzone

**Affiliations:** Department of Anatomy and Physiology, School of Biomedical Science, The University of Melbourne, Parkville, VIC 3052, Australia; stuart.mazzone@unimelb.edu.au

**Keywords:** vagal, hypersensitivity, airway innervation, development, sensory, children

## Abstract

A cough is one of several defensive responses that protect and clear the airways of inhaled, aspirated or locally produced chemicals and matter. The neural components needed to initiate a cough begin to develop in utero, and at birth the airways and lungs already have a rich supply of sensory and motor-neural innervation. However, a cough is not always the primary defensive response to airway challenge in very young infants, but instead develops in the first postnatal months and matures further into puberty. Consequently, the clinical presentation of a troublesome cough in children may not be the same as in adults, exemplified by important differences in cough sensitivity and hypersensitivity between children and adults. This review will summarise key anatomical and functional concepts in airway neurobiology that may improve understanding of coughs in children.

## 1. Introduction

A troublesome cough represents one of the most common reasons for people to seek medical advice. This prevalence of cough morbidity reflects the frequency of the cough as a presenting symptom across a range of acute and chronic conditions, including respiratory infections, exposure to smoke and other environmental irritants, obstructive and restrictive lung diseases, oesophageal reflux, nasal and sinus diseases, and lung cancer. Consequently, a cough can be troublesome at any age across a lifespan, periodically impacting individuals from birth to death. Much of what we understand about the cough has been derived from studies in adult animals and adult humans. However, the clinical picture of a troublesome cough in children is different to that in adults, suggesting that nuanced mechanisms may be at play [1]. These clinical considerations are discussed in detail in other contributions to this review series [2,3]. This review will explore the fundamentals of cough neurobiology to assess whether developmental differences in neural innervation or defensive-behaviour physiology may contribute to the clinical presentation of coughs in children.

## 2. Cough Phenomenology

The cough is a stereotyped, vagally-mediated respiratory response that is vitally important for airway clearance and the maintenance of airway patency [4]. In health, cough-evoking stimuli typically reach the larynx and airways via inhalation or aspiration of airborne chemicals or oro-pharyngeal materials which subsequently have the potential to obstruct or injure the airways. When triggered by these stimuli, a cough often occurs reflexively and explosively to rapidly expel the causative agent and, therefore, represents an important defence mechanism to protect the airways. In disease, a reflex cough can similarly be caused by inhaled or aspirated materials, but also by endogenous airway stimuli including excessive mucous or inflammatory-mediator activation of airway nerves. In addition, a cough may be evoked or facilitated by extra-pulmonary sites. For example, a reflex cough can be triggered from mechanical stimuli applied to the external auditory meatus through Arnold’s nerve reflex [5,6], involving the auricular branch of the vagus nerve. A cough may also be evoked or facilitated through stimuli applied to the oesophagus or nose, putatively through shared airway–oesophageal/nasal reflex pathways [4]. However, a cough can be more than a stereotyped reflex. Individuals can volitionally cough on command and display some level of conscious control over coughing which allows for the suppression or stifling of an evoked cough, as well as coughing more forcefully than would otherwise be expected [7,8,9]. This volitional control is accompanied by a perception of airway irritation, often felt as an itch or scratch in the throat, and clinically termed the urge to cough [10,11]. Furthermore, higher order cognitive processes can influence coughing, allowing for expectation and past experiences to influence behaviour, such as that seen with placebo cough suppression or anxiety-induced cough exacerbation [12,13,14]. Many of these processes become particularly important in a cough accompanying disease, changing the otherwise-defensive reflex response into a troublesome more complex multidimensional behaviour [4].

## 3. Basics of Cough Neurobiology

The cough reflex involves a neural circuit beginning with sensory neurons derived from the vagus nerves, which innervate the airway mucosa and project centrally to the brainstem [4]. Vagal sensory neurons reside within the jugular and nodose vagal sensory ganglia, which are paired structures that are located bilaterally at the cranial end of the nerve (reviewed in [15]). In animals, two types of cough-evoking sensory neurons have been identified, each with innervation patterns predominately in the large airways and larynx where a cough is often triggered (Figure 1). One type is derived from the jugular vagal ganglia and consists of peptidergic specialised chemosensory neurons, equipped with ion channels and receptors for a range of irritant chemicals. This stereotypically includes members of the transient receptor potential (TRP) family, including TRPV1, responsive to noxious heat, acid and capsaicin (from hot chillis); TRPA1, responsive to noxious cold temperature; and a wide range of pungent chemicals including allyl isothiocyanate (from wasabi), acrolein, aldehydes, eugenol, gingerol and cinnamaldehyde, to name a few [15]. Jugular cough sensory neurons are also responsive to a range of inflammatory mediators, including nicotine and other chemicals, making them analogous to the pain-evoking nociceptive nerve fibres that innervate the skin. The second type of cough-evoking sensory neurons are derived from the nodose vagal ganglia and are non-peptidergic specialised mechanosensory neurons, responsive to physical stimuli which distort the airway mucosa, such as inhaled or aspirated particulate matter or mucus [16]. Nodose cough sensory neurons are relatively ‘hard wired’ to evoke a cough, as even under anaesthesia their activation can initiate the reflex, perhaps reflecting their essential role in defending against catastrophic airway obstruction that could follow aspiration.

Cough-evoking sensory neurons send information to the brainstem, terminating in the medulla oblongata in two different locations, the nucleus of the solitary tract (nodose neurons) and the paratrigeminal nucleus (jugular neurons) [17,18]. Some outputs from these locations are sent to the brainstem’s respiratory circuits responsible for controlling rhythmic breathing, allowing the breathing pattern to be temporarily reconfigured into a cough motor pattern [19]. This is the neural basis for a reflex cough. Additionally, information is sent to complex higher-cortical brain networks, which allow for the urge to cough to be encoded and for higher brain volitional and cognitive controls to be overlaid onto the basic-reflex circuit (reviewed in detail here [4,15]). Whereas the neurobiology of a reflex cough has principally been studied in animals, the higher brain networks have largely been identified through brain-imaging studies in humans, including patients with a troublesome cough [9,10,20,21]. Indeed, central controls over the cough may be particularly relevant for paediatric populations, where somatic cough syndrome (formerly named psychogenic cough) is a common cause of troublesome coughing [22], as cough induction is likely driven by top-down (brain to brainstem) rather than bottom-up (airways to brainstem) processes in children with this condition.

## 4. Developmental Maturation of Airway Neural Innervation

The mature airways and lungs are densely innervated by sensory and motor (parasympathetic and sympathetic) nerve fibres, terminating onto structures in the mucosal, submucosal, muscle and adventitial layers [23,24,25,26]. The embryological and postnatal development of this neural innervation has been investigated in several species, including humans, although the level of description is perhaps not as complete as compared to other tissues, for example for the gastrointestinal system. In mice, pigs and humans, the development of the airways and lungs along with their innervation begins prenatally in the first trimester. By the end of this period, the airways possess a well-defined layer of smooth-muscle cells with an extensive nerve plexus that includes nerve trunks and intrinsic ganglia [27,28,29,30]. Electrical stimulation evokes neural-dependent airway-smooth-muscle contractions, suggesting early functionality in this innervation [27]. Soon after this, an extensive mucosal nerve plexus has formed, comprising distinct populations of sensory fibres to the epithelium and mucosal glands and sympathetic fibres to the developing mucosal vasculature, characterised by their expression of either the sensory neuropeptide calcitonin gene-related peptide (CGRP) or the precursor enzyme for noradrenalin synthesis tyrosine hydroxylase, respectively [27,31]. By late gestation in the pig, this mucosal plexus and the associated vasculature is well developed and by 4 weeks postnatal a dense CGRP fibre network is present in the airway epithelium, 2–35 μm below the luminal surface, comparable to that observed in adults. However, at birth, very few axons in the major vagal branches to the airways are myelinated, with this occurring progressively postnatally, peaking in adulthood [32]. Unfortunately, no studies have attempted to discriminate the development of specific sensory-nerve-fibre subtypes, beyond mapping CGRP-expressing fibres. As CGRP is preferentially expressed by jugular, and not nodose, vagal neurons, [33] it seems likely that the peripheral anatomical components of the jugular vagal cough pathway are established before, or soon after, birth. Even though the nodose cough fibres have not been specifically visualised during development, given the overall maturation state of airway innervation in early postnatal life, one could assume that nodose cough innervation to the airways is also established at this time.

## 5. Developmental Maturation of Airway Defensive Reflexes

As detailed above, vagal innervation to the airways is established early in prenatal development, and, at least for motor control of the airway smooth muscle in many mammals, there is well-developed functionality at the time of birth [34]. However, the early developmental presence of neural innervation in the airways does not necessarily mean that all airway neurobiology is established by birth or is indistinguishable from adult evoked responses. This may be especially true for reflexes and associated behaviours that require more intricate circuitry to become operationally refined. Indeed, a number of studies have shown that vagal reflexes are not static throughout life, but rather change in their physiological presentation from birth to old age [32,35,36]. Perhaps the best example of this is the Hering–Breuer stretch receptor reflex, activated by lung inflation, and involved in inspiratory termination in young mammals, but less so in adults [37]. In terms of airway defence, a cough has not been reported in a foetus, which may be a pragmatic developmental adjustment to reflex functionality, reflecting the fluid-filled lung environment during in utero life. This is not to say that prenatal infants do not have airway-defensive reflexes, but rather that the nature and purpose of these reflexes may change with development. Consistent with this, pre-term and full-term neonates have a strong laryngeal chemoreflex, in which apnoea is often the predominant respiratory response to water or other chemical stimuli applied to the glottis [35,38,39] (Figure 2). Apnoeic responses suppress inspiratory efforts presumably to limit aspiration, while swallowing then clears away the offending stimulus.

Although a cough may not be the most common component of the laryngeal chemoreflex in experimental studies of newborns, it is occasionally evoked, perhaps dependent on the level of sedation. For example, in non-sedated, sleeping full-term lambs, water and acid solutions applied to the glottis 2–3 days after birth resulted in defensive responses more consistent with lower airway-protective reflexes, including coughs, rather than the predominate apnoea and swallow as seen in sedated or pre-term animals [38]. Indeed, cough-reflex testing has not been conducted in very young infants and as such the developmental-related propensity of neonates to respond by coughing to airway challenge has not been systematically investigated. It is notable, however, that a cough can be clinically present, accompanying airway pathologies, in very young infants. Moreover, coughs in response to laryngeal stimulation in experimental studies becomes increasingly prevalent with advancing age in children (Figure 2). For example, in humans, a cough can be present as part of the laryngeal chemoreflex or evoked with mechanical stimulation in preterm infants, but it becomes more readily evoked after 1 month in full-term infants and during the first year of postnatal life [35]. In addition, although the clinical cough can be associated with infectious disease very early in postnatal life, it is more common in older children.

Taken together, these data suggest an age-related maturation of cough neurobiology beginning from birth, rather than a categorical switch from apnoea to cough at some point in development. This maturation is typified by the laryngeal chemoreflex, and presents as an age-dependent increase in the likelihood of cough induction over swallowing and apnoea upon airway challenge. The mechanisms of this developmental transition are not clear but may involve changes in the central processing of airway sensory inputs rather than alterations to peripheral sensory nerve terminal sensitivity or innervation patterns in airway tissues [32]. It is interesting that the Hering–Breuer reflex declines with age in childhood as activation of the lung stretch receptors mediating this reflex can inhibit coughs [40]. Although not known for certain, it seems likely that the centrally-mediated behavioural and cognitive controls over coughs that become clinically important in adults, also undergo concomitant and parallel postnatal development and help to shape cough maturation. In addition, significant changes to central-breathing-control systems, anatomical developments of the soft palate accompanying vocalisation, and generalised maturation of the nervous system all occur [41,42], and each could contribute to cough maturation. Regardless, most children have a cough reflex from early in childhood, even if it remains unclear what physiological factors ultimately determine the nature of the evoked reflex behaviour (cough or apnoea) that accompanies airway challenge during postnatal development.

## 6. Cough Hypersensitivity and Children

In adults, acute and chronic cough are both associated with a change in the sensitivity of cough induction by peripheral triggers (reviewed in [4]). This hypersensitivity is presumed to be a result of altered cough-pathway responsivity due to airway inflammation lowering cough-sensory-neuron activation thresholds and a vagal neuropathy or central (brain) mechanisms leading to amplification of incoming cough signals. In addition, cough induction can be facilitated by activation of sensory nerves innervating the oesophagus and nose. This latter mode of cough hypersensitivity is thought to be important in patients who have oesophagitis or rhinitis, and this mechanism of sensitization probably arises due to convergent interactions between the airway and oesophageal/nasal sensory neuron inputs at overlapping brainstem processing sites [4]. Consequently, patients cough in response to relatively low levels of thermal, chemical, or mechanical stimuli, a condition commonly referred to as cough hypersensitivity syndrome [43]. Clinically, cough hypersensitivity syndrome presents as a cough triggered by otherwise innocuous changes in air temperature, smells (e.g., perfumes), laughing, speaking on the phone or other everyday exposures, but additionally stimuli applied to non-airway territories of the vagus, such as the external ear, are also more likely to trigger coughing (Arnold’s nerve reflex) in adults [5,6]. Cough sensitivity in adults may also be defined by biological sex [44]. Adult females have a more sensitive cough reflex than do adult males, as demonstrated by cough challenge testing with inhaled tussive stimuli, such as capsaicin. The reason for this is unclear, although the heightened responsivity in females has been proposed to help protect against aspiration during pregnancy [45]. Alternatively, females also have lower pain reflex thresholds [46] suggesting that noxious sensory processing in general is different between adult males and females.

The reason cough hypersensitivity has become an important overarching concept in the field is because it now serves as a driver for advancements in therapeutic management and drug development [4]. Indeed, targeting hypersensitivity is thought to be the best strategy for treating a troublesome cough rather than blocking a cough entirely. However, it is debateable whether cough hypersensitivity is an important mechanism in coughing in children [2]. In one study, although the prevalence of Arnold’s nerve cough reflex was shown to be 11-fold higher in adults with a chronic cough compared to healthy adults, children displayed a very low prevalence of this reflex regardless of whether they were chronic coughers or not [6]. Children do not display sex differences in cough-challenge test sensitivity, which may be evidence that the mechanisms predisposing a hypersensitivity phenotype have not yet developed in children [47] (Figure 2). Furthermore, although children can have oesophageal and nasal diseases, it has not been investigated if activation of sensory nerves innervating these tissues can facilitate a cough, as has been shown in adults. Consequently, alternative frameworks for considering mechanisms and managing coughs in children are necessary (reviewed in this series in [2]).

It is intriguing that cough hypersensitivity may not be common in children given the commonalities in diseases that precipitate a cough in children and adults. Pulmonary inflammation accompanying acute airway infections and chronic airways disease (such as asthma) are believed to impact cough-evoking sensory nerve fibres in both children and adults, but perhaps this inflammation only serves to activate cough fibres in children, rather than contribute to hypersensitivity. The reasons for this remain speculative but could involve some resistance against the development of cough (vagal or central) neuropathy in early age. Consistent with this, children rarely present with neuropathic pain, for example associated with diabetes, postherpetic neuralgia, trigeminal neuralgia or radiculopathies [48].

## 7. Conclusions

The rich neural supply to the airways, whilst laid down in prenatal development, is not static in structure or function throughout an individual’s lifespan. Postnatal maturation of innervation patterns and neural circuits manifests as age-related changes in airway neurobiology, and this likely has important impacts on the clinical presentation of the cough in patients of different age demographics. Although we understand this in concept, there remains many gaps in our knowledge of the specific mechanisms underpinning developmental changes in coughs. In this regard, it will also be important to consider the impacts of early-life disease, environmental exposures, and genetic variances on the maturation of the airway nervous system, as developmental plasticity may result in important clinical consequences in terms of cough neurobiology and airway defence, lasting into adult life.

## Figures and Tables

**Figure 1 jcm-12-07285-f001:**
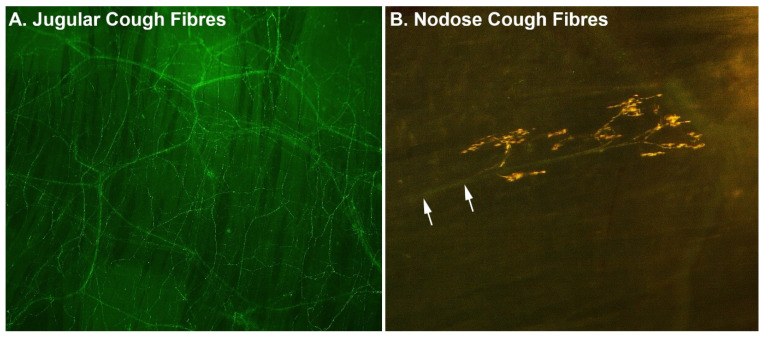
Examples of (**A**) jugular and (**B**) nodose cough-evoking nerve fibres in the trachea. Images show representative stains of nerve fibres viewed from the luminal surface in tissue wholemounts collected from guinea pigs. Jugular cough neurons form a dense subepithelial plexus, with fibre branches extending between epithelial cells all the way to the luminal surface. By contrast, nodose cough mechanosensors are sparse and characteristically display a distinct parent axon (arrows), which ramifies into a complex subepithelial nerve ending. Comparable structures have been identified in human airways.

**Figure 2 jcm-12-07285-f002:**
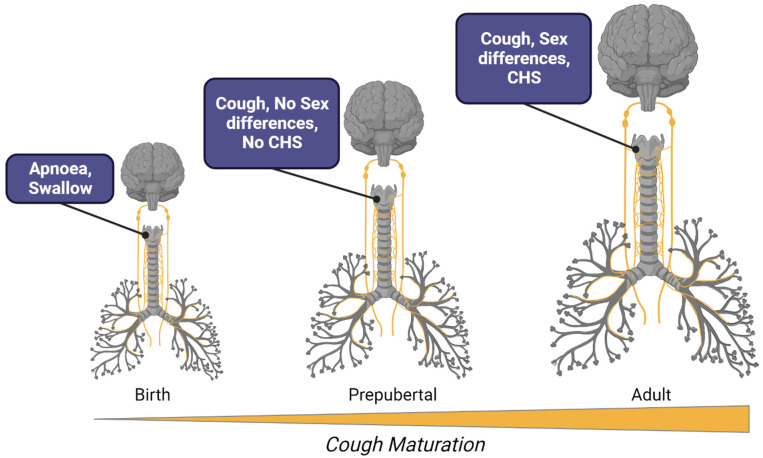
Schematic representation of major developmental milestones in cough neurobiology. At birth, the airways are densely innervated by all nerve-fibre types. Laryngeal stimuli in the newborn predominately evokes the laryngeal chemoreflex, characterised by a cessation of breathing and swallowing. Apnoea is replaced by a cough later in postnatal development, but in prepubescent children the cough reflex has yet to fully mature, lacking the well-known sex differences in reflex sensitivity and an absence of cough hypersensitivity (CHS) as a major component of clinical cough in disease. By adulthood, females demonstrate a heighten reflex sensitivity compared to males and both females and males commonly display CHS in clinical conditions leading to an acute and chronic cough.

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
