# Peer review of "Neurobiology of Coughing in Children"

_jcm, 2023, doi:10.3390/jcm12237285_

Round 1

Reviewer 1 Report

Comments and Suggestions for Authors

I enjoyed reading this review. It is well written. My major issue is that cough is viewed as arising exclusively from the airways. It is increasingly recognised that chronic cough is a disease of the entire vagus and its central projections. There is no mention of the heavy innervation of the oesophagus. Oesophageal disease as a cause of cough is almost completely ignored by paediatricians. Surely the gut lung axis at least deserves a paragraph.

Infants reflux frequently, there is even a word for it – positing, but are protected against aspiration because they have an intact valve mechanism consisting of the arytenoid cartilage, epiglottis, and the soft palate (as in all other mammals). When the child begins to speak the soft palate migrates cranially to allow for vocalisation.  Laitman and Reidenberg Am J Med 1997. Perhaps this is why apnoea is a common response in infants.

It is incorrect to state that cough is not a primary defensive response to airway challenge in very young infants. An example is babies who are born with tracheal oesophageal fistula. The fistula is usually repaired in the first few days following birth, but the infant coughs even after the anatomical repair – so-called TOF cough. Whilst the tubes are connected the neuroanatomy is a disordered leading to oesophageal dysmotility.

There are many more agonists of TRPA1 please expand.

Somatic cough syndrome (27) is really a forme fruste of Tourette’s.1 Even though my colleagues disagree. It is important here because it shows chronic cough in children can be driven by central mechanisms and is not necessarily volitional.

Girls develop a heightened cough reflex during puberty. Rather than saying that the reason is unclear perhaps say it has been speculated that this is to protect against aspiration during pregnancy.

The most interesting thing about Peter’s study (46) is the enormous difference between adults and children rather than children with cough versus non-coughers. The numbers are less than a handful.

1.            Ojoo JC, Kastelik JA and Morice AH. A boy with a disabling cough. Lancet 2003; 361: 674-674.

Author Response

1. I enjoyed reading this review. It is well written. My major issue is that cough is viewed as arising exclusively from the airways. It is increasingly recognised that chronic cough is a disease of the entire vagus and its central projections. There is no mention of the heavy innervation of the oesophagus. Oesophageal disease as a cause of cough is almost completely ignored by paediatricians. Surely the gut lung axis at least deserves a paragraph.

Response: I have now included discussion of this topic in relevant places in the manuscript.

2. Infants reflux frequently, there is even a word for it – positing, but are protected against aspiration because they have an intact valve mechanism consisting of the arytenoid cartilage, epiglottis, and the soft palate (as in all other mammals). When the child begins to speak the soft palate migrates cranially to allow for vocalisation.  Laitman and Reidenberg Am J Med 1997. Perhaps this is why apnoea is a common response in infants.

Response: Thank you for this reference.  I have now included soft palate maturation as a possible contributor to cough maturation and cited Laitman’s work.

3. It is incorrect to state that cough is not a primary defensive response to airway challenge in very young infants. An example is babies who are born with tracheal oesophageal fistula. The fistula is usually repaired in the first few days following birth, but the infant coughs even after the anatomical repair – so-called TOF cough. Whilst the tubes are connected the neuroanatomy is a disordered leading to oesophageal dysmotility.

Response: I appreciate the comment and can see why you have raised this concern. I didn’t mean to suggest that cough is not possible from birth, but rather to paint a picture of developmental maturation whereby coughing (in response to airway challenge) becomes more probable with advancing age in early postnatal development.  I have amended this section to better reflect this and mention that pathological cough does occur clinically at young ages.

4. There are many more agonists of TRPA1 please expand.

Response: The list has been expanded a little, but the purpose is not to be exhaustive as this is not a key focus of the manuscript.

5. Somatic cough syndrome (27) is really a forme fruste of Tourette’s.1 Even though my colleagues disagree. It is important here because it shows chronic cough in children can be driven by central mechanisms and is not necessarily volitional. 

Response: I am aware of the conjecture surrounding this topic and take on board your point that the mechanisms in somatic cough need not be ‘volitional’.  Rather than reopen the debate on somatic cough and Tourette’s, (which has been heated at times!), I have edited the section to better represent the likely central origins of cough in Somatic cough syndrome.   

6. Girls develop a heightened cough reflex during puberty. Rather than saying that the reason is unclear perhaps say it has been speculated that this is to protect against aspiration during pregnancy.

Response: I have added this speculation.

7. The most interesting thing about Peter’s study (46) is the enormous difference between adults and children rather than children with cough versus non-coughers. The numbers are less than a handful.

Response: Agreed.  The sentence has been edited to convey this more clearly.

Reviewer 2 Report

Comments and Suggestions for Authors

Stuart B. Mazzone presented a narrative review covering the topic of the Neurobiology of cough in children.

Cough is a major symptom of daily clinical practice regardless of age. Every knowledge highlighting the mechanisms of cough is essential and has a significant clinical value. 

The Mazzone review is informative. However, there are issues to be addressed:

1.     The text is not formatted in the Journal Template, which is highly recommended. It will help the author structure the text as a narrative review. In the present form, the text looks more like a notebook chapter. 

2.     Introduction is missing

3.     Figures 1 and 2 have no explanation.

4.     Figure 1 – Does the author own the copyright? Is it human or animal tissue?

5.     More detailed and comprehensive reviews on that topic have already been published in the last two years (Al-Biltagi M, Bediwy AS, Saeed NK. Cough as a neurological sign: What a clinician should know. World J Crit Care Med. 2022 May 9;11(3):115-128. doi: 10.5492/wjccm.v11.i3.115. PMID: 36331984; PMCID: PMC9136724.) I recommend the author highlight the differences between children and adults in each review section. 

Author Response

1. The text is not formatted in the Journal Template, which is highly recommended. It will help the author structure the text as a narrative review. In the present form, the text looks more like a notebook chapter. 

Response: The paper is structured as a review paper, which is what was requested from the guest editors. As such, the subheadings define the areas of content covered. It makes little sense to use the template which is in IMRAD format.    

2. Introduction is missing

Response: A brief introduction is now included.

3. Figures 1 and 2 have no explanation.

Response: The reviewer has missed these as the legends were included in the original submission.

4. Figure 1 – Does the author own the copyright? Is it human or animal tissue?

Response: Refer to figure legends which contain the details.

5. More detailed and comprehensive reviews on that topic have already been published in the last two years (Al-Biltagi M, Bediwy AS, Saeed NK. Cough as a neurological sign: What a clinician should know. World J Crit Care Med. 2022 May 9;11(3):115-128. doi: 10.5492/wjccm.v11.i3.115. PMID: 36331984; PMCID: PMC9136724.) I recommend the author highlight the differences between children and adults in each review section.

Response: The review by Al-Biltagi and colleagues is focused on neurological disorders that may have cough as an accompanying symptom. Many of the disorders presented are obscure causes of cough.  The neurobiology presented in their review is scant and not always correct.  It is unclear how the paper relates to the current submission.  Relevant comparisons to adults have been made throughout all sections of the manuscript.

Reviewer 3 Report

Comments and Suggestions for Authors

The author presents an interesting review of the neurobiological mechanisms ivovled in cough in children.

However, very little is debated on pathological cough, which is the most common cause of cough in children, eg. in asthma (since asthma is the most common chronic disease in the pediatric population). There is evidence that cough in asthma is linked to neuronal dysfunction exacerbated by the underlying inflammation, and it would be good to review the pathophysiology of cough in this manuscript, too.

Author Response

However, very little is debated on pathological cough, which is the most common cause of cough in children, eg. in asthma (since asthma is the most common chronic disease in the pediatric population). There is evidence that cough in asthma is linked to neuronal dysfunction exacerbated by the underlying inflammation, and it would be good to review the pathophysiology of cough in this manuscript, too.

Response: I have added a little more on this topic.  The notion that inflammation causes neuronal dysfunction is appealing, but the evidence is largely based on animal studies in which the models of asthma are less than ideal.  This is not to say that inflammation is not important for modulating neural function, but rather that there is a lack of high-quality data in the clinical setting to allow for specific commentary, and very little in children.  Expert opinion on the role of inflammation in activating sensory nerves and causing chronic cough is divided, with major reservations around this presumption (PMID: 25142479).  The author has also written recently on airway neuro-immune interactions (PMID: 37517657).  For these reasons, I’ve kept the discussion appropriately brief and focused on the neurobiology as requested by the guest editors.

Reviewer 4 Report

Comments and Suggestions for Authors

The review "Neurobiology of cough in children” highlights the role of anatomical and functional differences in understanding cough in children. This was interesting, but there are some problems regarding the structure and content.

I suggest a few major revisions. Comments are made below regarding the article.

1. The authors should add an introduction with the aim and objectives. 2. The Methodology should be numbered. (1. Cough phenomenology

2. Developmental maturation of airway neural innervation, eg).

3. Since inflammation is the main important cause of cough in children, should be added in methodology and discuss the impact of inflammation on the nerves and the interaction between the immune system and the nerves in the airways.

4. Add discussion if it is possible to explain the differences and the clinical impact of neurobiology of cough as other authors observed in their studies.

Comments on the Quality of English Language

English language should be verify by a specialist.

Author Response

1. The authors should add an introduction with the aim and objectives. 2. The Methodology should be numbered. (1. Cough phenomenology).

Response: A brief introduction has been included and the editorial-level corrections made.

2. Developmental maturation of airway neural innervation, eg).

Response: Unable to address this comment as the sentence is incomplete.

3. Since inflammation is the main important cause of cough in children, should be added in methodology and discuss the impact of inflammation on the nerves and the interaction between the immune system and the nerves in the airways. I have added a little more on this topic.

Response: The notion that inflammation causes neuronal dysfunction is appealing, but the evidence is largely based on animal studies in which the models of asthma are less than ideal. This is not to say that inflammation is not important for modulating neural function, but rather that there is a lack of high-quality data in the clinical setting to allow for specific commentary, and very little in children.  Expert opinion on the role of inflammation in activating sensory nerves and causing chronic cough is divided, with major reservations around this presumption (PMID: 25142479).  The author has also written recently on airway neuro-immune interactions (PMID: 37517657).  For these reasons, I’ve kept the discussion appropriately brief and focused on the neurobiology as requested by the guest editors.

4. Add discussion if it is possible to explain the differences and the clinical impact of neurobiology of cough as other authors observed in their studies.

Response: It is unclear what point the reviewer is trying to make. As this is an optional edit and given the lack of clarity, I have chosen not to add such a discussion.  This comment may be satisfied by knowing that this paper is part of a series, others of which will deal with clinical issues in paediatric cough.

Round 2

Reviewer 2 Report

Comments and Suggestions for Authors

The manuscript is significantly improved. 

Author Response

There doesn't appear to be any comments to address.

Reviewer 4 Report

Comments and Suggestions for Authors

No other comments

Comments on the Quality of English Language

No other comments

Author Response

(The authors gave the same response as above.)
